# The role of migration networks in the development of Botswana's generalized HIV epidemic

Janet Song[1], Justin T Okano[1], Joan Ponce[1], Lesego Busang[2], Khumo Seipone[2], Eugenio Valdano[3], Sally Blower[1]*

[1]Center for Biomedical Modeling, Department of Psychiatry and Biobehavioral Sciences, Semel Institute for Neuroscience and Human Behavior, David Geffen School of Medicine, University of California, Los Angeles, Los Angeles, United States; [2]The African Comprehensive HIV/AIDS Partnerships (ACHAP), Gaborone, Botswana; [3]Sorbonne Université, INSERM, Institut Pierre Louis d'Epidémiologie et de Santé Publique, Paris, France

**Abstract** The majority of people with HIV live in sub-Saharan Africa, where epidemics are generalized. For these epidemics to develop, populations need to be mobile. However, the role of population-level mobility in the development of generalized HIV epidemics has not been studied. Here we do so by studying historical migration data from Botswana, which has one of the most severe generalized HIV epidemics worldwide; HIV prevalence was 21% in 2021. The country reported its first AIDS case in 1985 when it began to rapidly urbanize. We hypothesize that, during the development of Botswana's epidemic, the population was extremely mobile and the country was highly connected by substantial migratory flows. We test this mobility hypothesis by conducting a network analysis using a historical time series (1981–2011) of micro-census data from Botswana. Our results support our hypothesis. We found complex migration networks with very high rates of rural-to-urban, and urban-to-rural, migration: 10% of the population moved annually. Mining towns (where AIDS cases were first reported, and risk behavior was high) were important in-flow and out-flow migration hubs, suggesting that they functioned as 'core groups' for HIV transmission and dissemination. Migration networks could have dispersed HIV throughout Botswana and generated the current hyperendemic epidemic.

*For correspondence:
sblower@mednet.ucla.edu

Competing interest: The authors declare that no competing interests exist.

## Editor's evaluation

This valuable paper uses representative samples of micro-census data from Botswana to describe migration rates over four points in time, from 1981 to 2011. The authors use compelling descriptive data to present migration characteristics where roughly 10% of the population moved in the past year – with equal numbers of men and women, and with migration between districts more common than within districts. Preliminary data indicated migration patterns could have supported HIV diffusion, this can be a starting point for more in-depth analyses. The work will be of interest to those studying human movement and its impact on diseases.

## Introduction

Over 25 million people live with HIV infection in sub-Saharan Africa. All of the HIV epidemics in this continent are generalized: in these types of epidemics, the epidemic is dispersed throughout the country. Therefore, a population needs to be highly mobile in order for a generalized epidemic to

**eLife digest** Over 25 million people in sub-Saharan Africa live with HIV. After reporting its first AIDS case in 1985, Botswana is one of the most severely affected countries in the region, with one in five adults now living with HIV.

Movement of the population is likely to have contributed to a geographically dispersed, and high-prevalence, HIV epidemic in Botswana. Since 1985, urbanization, rapid economic and population growth, and migration have transformed Botswana. Yet, few studies have analyzed the role of population-level movement patterns in the spread of HIV during this time.

By studying micro-census data from Botswana between 1981 and 2011, Song et al. found that the country's population was highly mobile during this period. Reconstructions of internal migration patterns show very high rates of rural-to-urban and urban-to-rural migration, with 10% of Botswana's population moving each year. The first reported AIDS cases in Botswana occurred in mining towns and cities where high-risk behavior was prevalent. These areas were also migration hubs during this period and could have contributed to the rapid spread of HIV throughout the country as infected individuals moved back to rural districts.

Understanding human migration patterns and how they affect the spread of infectious diseases using current data could help public health authorities in Botswana and additional sub-Saharan African countries design control strategies for HIV and other important infections that occur in the region.

have developed. However, although the epidemiology of HIV in sub-Saharan Africa has been widely studied (*Farley et al., 2022*; *Read et al., 2022*), the role of population-level mobility patterns in the development of generalized HIV epidemics has not been assessed for any country on the continent. Botswana has one of the most severe HIV epidemics worldwide and reported its first AIDS case in the early 1980s. Since then, the epidemic has become generalized and hyperendemic: in 2021, HIV prevalence in adults (15–64) was 21% (*Mine et al., 2022*). Simultaneous to the development of its generalized HIV epidemic, Botswana has undergone rapid urbanization. It was predominantly rural in 1981, but had become predominantly urban by 2011 (*Statistics Botswana, 2014*); by that time Botswana's HIV epidemic had stabilized (*CSO Botswana, 2009*). We hypothesize that – during the development of Botswana's epidemic – the population was extremely mobile and highly connected by substantial urban-to-rural and rural-to-urban migratory flows. We test this mobility hypothesis by conducting a network analysis using a historical time series of micro-census data collected in Botswana. These data contain information both on migration and urbanization. The time series covers the time period from when the epidemic was first apparent to when it stabilized in 2011. We use these data to (i) estimate the annual incidence (at the national level) of internal migration over three decades (1981–2011), (ii) characterize migrants on the basis of gender and age, (iii) reconstruct internal migration networks (in order to identify large-scale population movements and connectivity patterns), (iv) identify migration hubs, (v) understand and visualize the role of migration networks in urbanization, and (vi) determine the extent to which internal migration led to a substantial geographic redistribution of the population. The Government of Botswana defines internal migration as residents changing their place of permanent residence within their home country. Finally, we discuss how the historical migration networks – that we identify in our analyses – had the potential to disperse HIV throughout Botswana and generate a hyperendemic epidemic.

After 80 y of colonial rule, Botswana gained independence from the United Kingdom in 1966 and has since been Africa's longest uninterrupted democracy. One of the poorest countries in the world at the time of independence, Botswana has since undergone rapid urbanization and economic expansion; the diamond mining industry is the economic foundation of Botswana (*Barnett et al., 2002*). Rich diamond deposits were discovered immediately after independence; the first diamond mine (Orapa) became fully operational in July 1971, the second mine (Jwaneng) in August 1982. These two mines are now amongst the world's richest diamond mines. Today, Botswana is an upper-middle-income country with a high demographic growth rate: the population approximately doubled from 1 to 2 million, from 1981 and 2011 (*Statistics Botswana, 2014*).

In 1981, the vast majority of the population lived in small rural villages. Only ~18% of the population were living in urban areas (*Statistics Botswana, 2014*): either in one of the two cities (Gaborone,

the capital, or Francistown) or in one of the four towns (Lobatse, Selebi Phikwe, Orapa, and Jwaneng). Lobatse is an administrative center, and the other three are mining towns. Selebi Phikwe is based on copper and nickel mining, and was founded in the early 1970s. By 2011, only ~36% of the population were living in rural areas (*Statistics Botswana, 2014*). The remaining population were living in one of the three types of urban centers: a city (there were still only two cities), a town (there were now five towns following the 1991 addition of Sowa, a mining town for soda ash), or an urban village. In Botswana, an urban village is defined as a settlement with at least 5,000 individuals and 75% of the workforce engaged in non-agricultural economic activities (*Statistics Botswana, 2014*). The urban villages developed by in situ urbanization, which is defined as rural settlements transforming into urban areas by expanding their non-agricultural activities and increasing economic linkages with neighboring areas (*Moriconi-Ebrard et al., 2020*). Botswana now consists of 28 administrative districts (*Okano et al., 2021*). Each city and town are separate administrative districts, and the remaining 21 districts each contain at least one urban village and many rural villages.

A great deal is known about the epidemiology of HIV in Botswana in terms of the temporal increase, and geographic variation, in prevalence (*Magosi et al., 2022*; *Novitsky et al., 2015*; *Novitsky et al., 2020*; *Okano et al., 2021*). HIV prevalence rose quickly and spread fairly widely. The first case of AIDS was reported in 1985 from the mining town of Selebi Phikwe (*African Natural Resources Center, 2016*) in the late 1980s; additional cases were reported from the two diamond mining towns (Jwaneng and Orapa) and the city, Francistown. In 1990, the first HIV Sentinel Surveillance Survey of antenatal clinic (ANC) attendees (15–49years old) was conducted in Gaborone and in one rural district (Boteti): HIV prevalence was found to be 6% and 4%, respectively (*UNAIDS and World Health Organization, 2004*). In 1992, a national sentinel surveillance survey began and was conducted annually to 2011. National prevalence in pregnant women was found to have already reached a very high level (18%) by 1992 and continued to increase for the next 8 y: 23% by 1993, 32% by 1995, and 39% by 2000 (*UNAIDS and World Health Organization, 2004*). Based on the ANC data, HIV prevalence in the cities and the mines was the highest. In Francistown, prevalence was 8% in 1991, 24% by 1992, and 44% by 2000. In Gaborone, prevalence rose from 15% in 1992 to 36% by 2000. In the mining town of Selebi Phikwe, prevalence rose from 27% in 1994 to 50% by 2000. By the late 1990s, a high AIDS morbidity and mortality rate in miners led the diamond mining company Debswana to conduct an HIV prevalence survey of its (male) employees. The survey took place in 1999: 29% of miners were found to be infected with HIV (*Barnett et al., 2002*). The first population-level survey in Botswana to test participants for HIV (BAIS II, the Botswana AIDS Impact Survey) was conducted in 2004 (*NACA and CSO Botswana, 2005*). It found, at that time, that HIV prevalence in the general population was 29% in women and 20% in men, aged 15–49 years old. Gender-stratified prevalence was the same in 2008, when BAIS III was conducted; the considerable geographic variation in HIV prevalence amongst districts that existed at that time is shown in *Figure 1*. Notably, phylogenetic studies have shown that viral lineages have dispersed widely throughout Botswana, suggesting that large-scale population-level movements have occurred (*Magosi et al., 2022*; *Novitsky et al., 2015*; *Novitsky et al., 2020*).

## Results
### Estimated incidence of internal migration
We estimated the Crude Migration Intensity (CMI) (*Bell et al., 2002*) for Botswana in the 12 mo before each census was conducted (i.e., between 1980 and 1981, 1990 and 1991, 2000 and 2001, and 2010 and 2011). The CMI represents the overall incidence, or level of internal migration, per hundred residents over a specified time interval (*Bell et al., 2002*); it is an indicator of the propensity of the population to move and is a measure of migration both between districts and within districts. The CMI remained markedly high and constant: 9.04 per hundred persons between 1980 and 1981, 10.45 per hundred persons between 1990 and 1991, 10.73 per hundred persons between 2000 and 2001, and 10.32 per hundred persons between 2010 and 2011. We found that migrants were more likely to move between districts than to move within districts; this propensity, calculated as a ratio (the number of migrants moving between districts relative to the number of migrants moving within districts), increased over time from 1.25 (1981) to 2.40 (2001 and 2011).

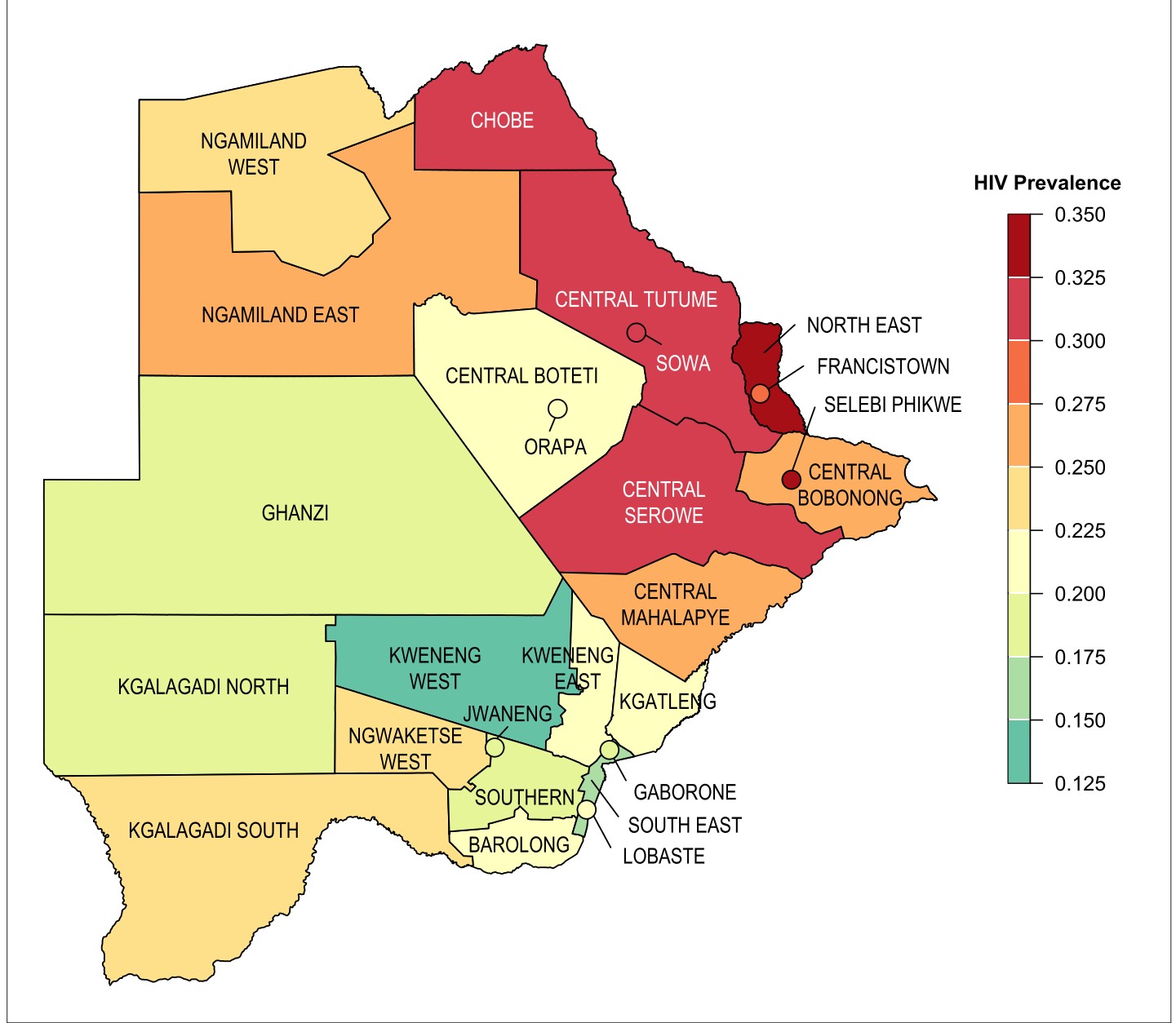

**Figure 1.** HIV prevalence at the district-level. HIV prevalence estimated using BAIS III data (2008) for adults ages 15–69. Botswana's two cities and five towns are denoted with circles. Districts for the Okavango Delta and Central Kgalagadi Game Reserve have been geographically incorporated into Ngamiland West and Ghanzi, respectively, due to their small population sizes.

## Age profiles of migrants

Gender-stratified age profiles of migrants are presented in *Figure 2*. These results show that the type of individual who migrated within Botswana at the time of each census was markedly similar, with respect to gender and age. Approximately 50% of migrants were women, and the most common age to migrate (for both women and men) was between 16 and 20 years old.

## Migratory flows, district size, and migration metrics

To reconstruct the internal migration networks, we first calculated migratory flows between districts. We define a district-level migratory flow as the number of migrants who changed their residency from one district to another during the 12 mo prior to the census. The migratory flows for each district (within district, in-flow, and out-flow) and the size of each district are listed for 1980–1981 (*Supplementary*

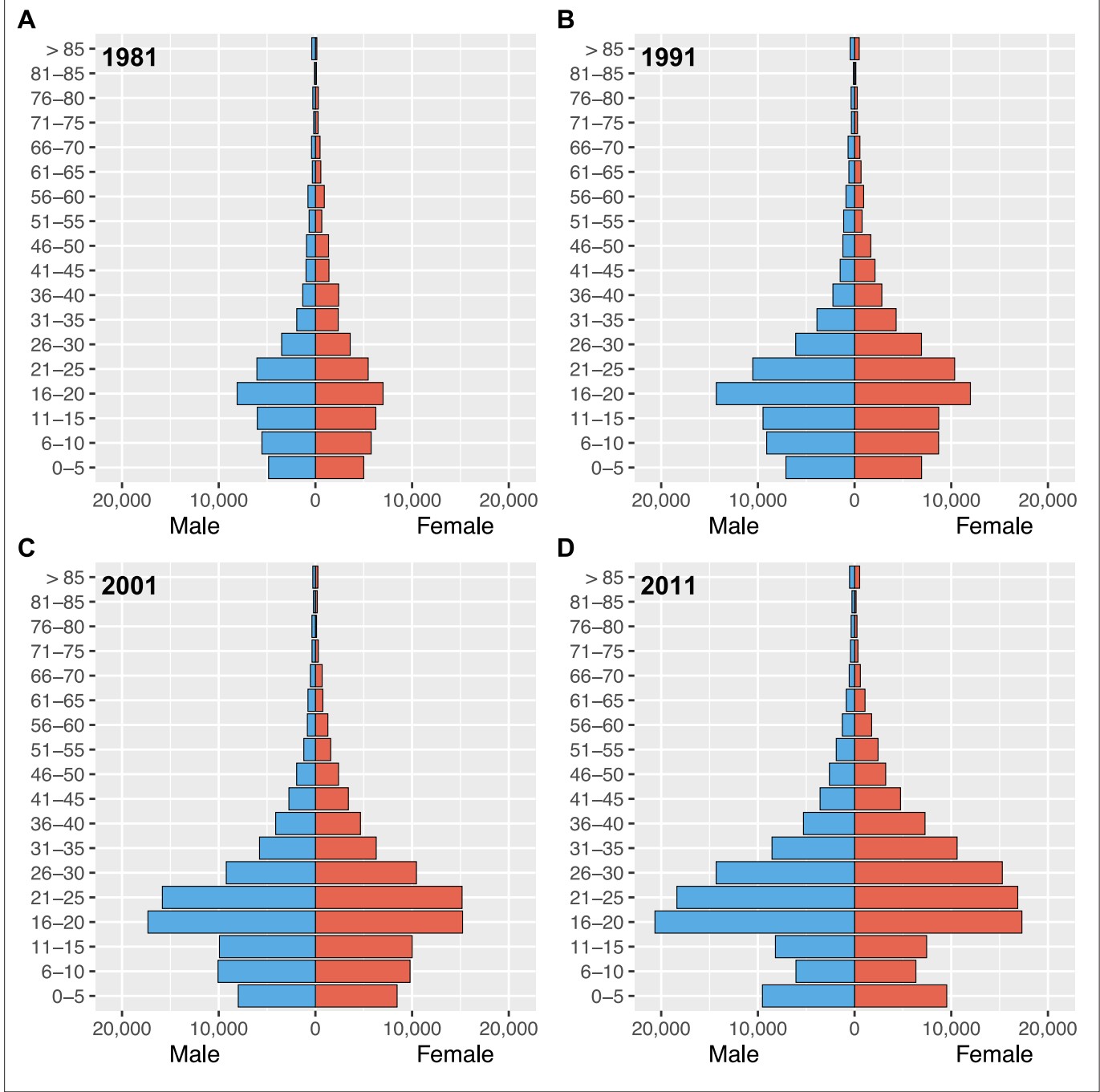

**Figure 2.** Gender-stratified age profiles of internal migrants by census year. (**A**) 1981, (**B**) 1991, (**C**) 2001, and (**D**) 2011.

file 1a), 1990–1991 (*Supplementary file 1b*), 2000–2001 (*Supplementary file 1c*), and 2010–2011 (*Supplementary file 1d*). All of the districts had an in-flow, and/or out-flow, of migrants at all four census periods (*Figure 3* and *Supplementary file 1*). Over the three decades (i.e., between 1981 and 2011), the number of migrants more than doubled, as did the population of Botswana.

Notably, our analysis of the micro-census data show that, in 1981, the population of the two cities and four towns were very small. The capital, Gaborone, had a population of 56,860 and Francistown had a population of only 31,120. The populations of the four towns ranged in size from 5,200 to 27,430. The number of individuals who lived in the other districts ranged in size from 1,050 in the Okavango Delta to 111,820 in Kweneng. In the early time periods, there was a large migration into many of the cities and towns; by 2011, there was a large migration out of many of them.

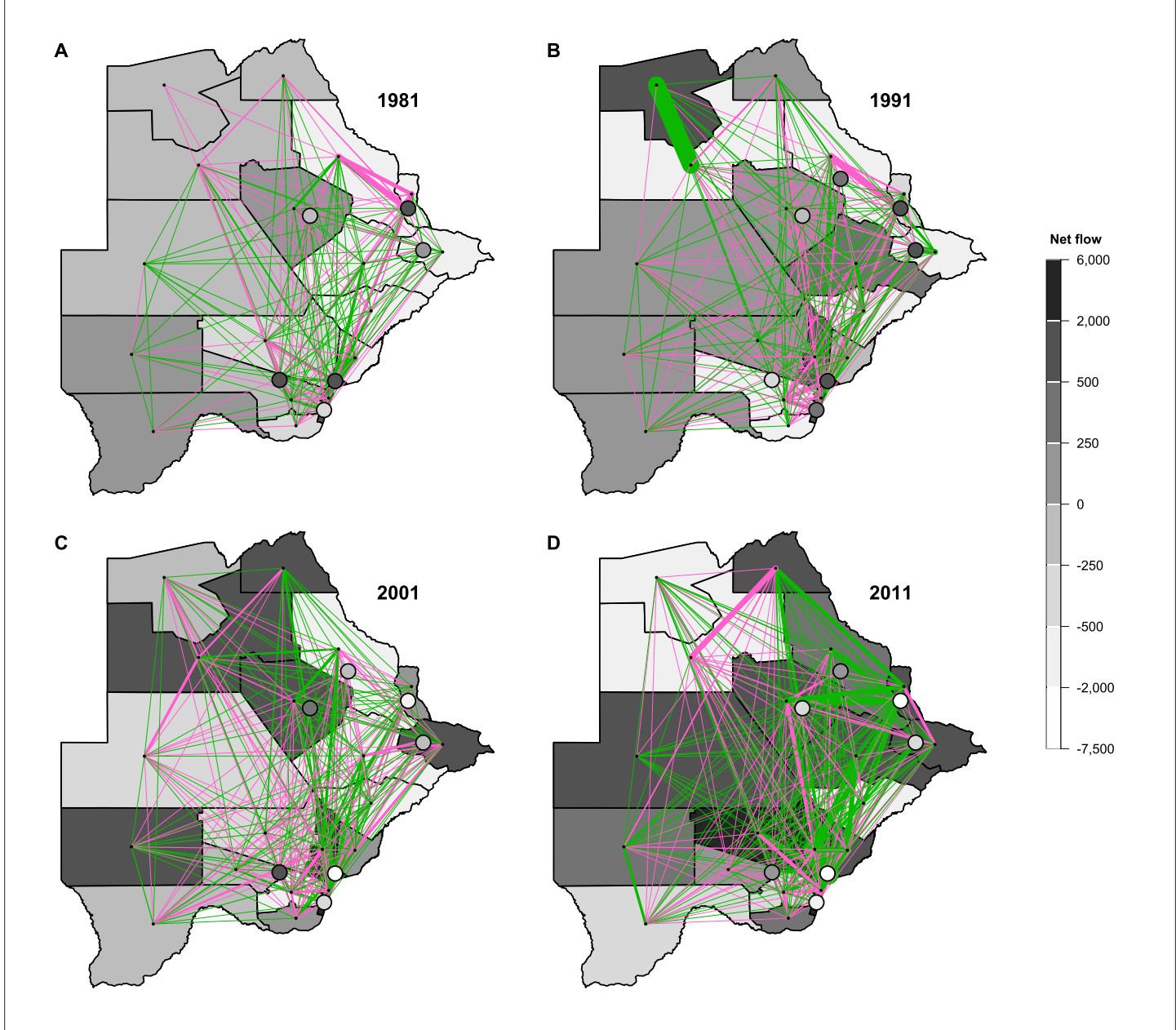

**Figure 3.** Maps of net migratory flows by census year. (**A**) 1981, (**B**) 1991, (**C**) 2001, and (**D**) 2011. Maps show the net migratory flows between any two districts (pink and green lines). Line thickness indicates the magnitude of the flow between districts, line color indicates the flow direction. Pink lines denote eastward flows, green lines denote westward flows. For example, in 1991, the thickest green line indicates a large (westward) migratory flow from Ngamiland East to Ngamiland West. Each district is shaded to indicate the total net flow of all migrations into and out of it. Cities and towns are represented with circles. Districts for the Okavango Delta and Central Kgalagadi Game Reserve have been geographically incorporated into Ngamiland West and Ghanzi, respectively, due to their small population sizes.

The online version of this article includes the following figure supplement(s) for figure 3:

**Figure supplement 1.** Maps of turnover by census year.

**Figure supplement 2.** Maps of within district migration intensity (WDMI) by census year.

Using the migratory flows, we estimated the annual turnover rate in each district in the year before each census (*Figure 3—figure supplement 1*). This rate is defined as the net change in the district's annual migration rate per hundred residents. A negative rate signifies that the district's population size decreased, a positive rate signifies that it increased. In 1981, many cities and towns had positive

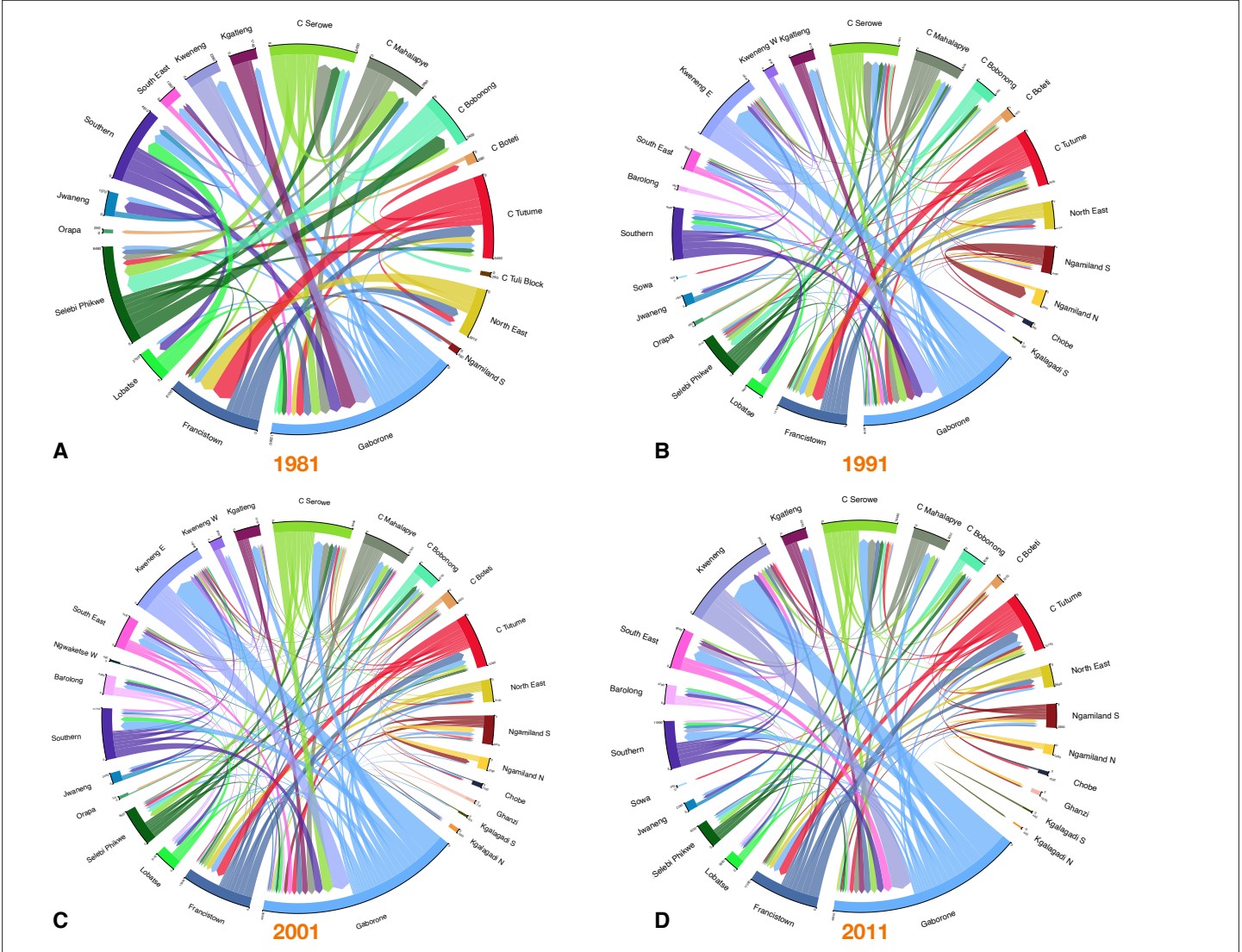

**Figure 4.** Chord diagrams by census year. (**A**) 1981, (**B**) 1991, (**C**) 2001, and (**D**) 2011. Each diagram shows the internal migration network of the general population in the 12 mo prior to the census. Each color represents a different district. The thickness of each line is proportional to the number of migrants that moved between the two connected districts. The angular width of each district is proportional to the total number of migrants who moved into, or out of, that district. For clarity, in (**A**–**C**) only connections with greater than 200 migrants are shown, and in (**D**) only connections with greater than 400 migrants are shown. Consequently, some districts are not shown in the chord diagram. The total number of migrants (in and out) of every district is listed in *Supplementary file 1*.

turnover, but by 2011, the majority had negative turnover. There were very high turnover rates in 1981, with the highest occurring in the mining town, Jwaneng, where turnover was ~45 per hundred residents.

More than half of the districts also had a fairly high within district migration intensity (WDMI) (*Figure 3—figure supplement 2*): for example, in Central Bobonong, ~7% of residents moved from one rural village to another between 1980 and 1981. The cities and towns have very low levels of within-district movement due to their small geographic size. In general, levels of WDMI were higher on the west side of the country, lower in the southeast, and fairly stable over time.

## Reconstructed internal migration networks

The chord diagrams (*Figure 4*) show the reconstructed internal migration networks, based on the micro-census data, between 1980–1981 (*Figure 4A*), 1990–1991 (*Figure 4B*), 2000–2001 (*Figure 4C*), and 2010–2011 (*Figure 4D*). Each district is a node in the network. Networks are shown in terms

of the magnitude of the migratory in-flows and out-flows between districts, and the effect of these flows on connecting different districts throughout the country: two districts are connected if they have a migratory flow between them. Even the earliest migration network in 1981 – when Botswana was predominantly rural – can be seen to be fairly complex and shows a high degree of connectivity amongst the districts, although the migratory flows and counter-flows are fairly small. Notably, in all four networks, many of the flows and counter-flows are similar in magnitude. Both the magnitude of the migratory flows and counter-flows increased with time (between 1981 and 2011) as the population increased in size.

## In-flow and out-flow migration hubs

The top five in-flow and out-flow migration hubs are shown in *Figure 5*. Of note, all five towns (Jwaneng, Selebi Phikwe, Sowa, Lobatse, and Orapa) were amongst the top five out-flow and in-flow migration hubs at every census period.

In 1981, the top in-flow hub was the diamond mining town, Jwaneng; 45% of the town's population consisted of migrants who had moved to the town in the previous 12 mo (*Figure 5A*). Jwaneng was also an important out-flow hub: 14% of the town's population moved to another district between 1980 and 1981 (*Figure 5B*). The ego networks for Jwaneng show the districts that migrants returned to (*Figure 6A*) and the districts that they came from (*Figure 6B*). It can be seen that migrants moved between the other four towns, the two cities, and many of the rural districts, with the majority of migrants moving between Jwaneng and Southern, a district abutting the town.

In 1991, 2001, and 2011, the top in-flow (and out-flow) hub was the mining town, Sowa. In 1991, 48% of the population of this town consisted of migrants who had moved to the town in the previous 12 mo (*Figure 5C*); in that same time period, 25% of the town's population moved to another district (*Figure 5D*). In 2001, 22% of the population consisted of migrants who had moved to the town in the previous 12 mo (*Figure 5E*); in that same time period, 25% of the town's population moved to another district (*Figure 5F*). In 2011, 24% of the population consisted of migrants who had moved to the town in the previous 12 mo (*Figure 5G*); in that same time period, 18% of the town's population moved to another district (*Figure 5H*).

Notably, Francistown was an important in-flow and out-flow hub between 1980 and 1981. In 1981, 15% of the population consisted of migrants who had moved to the town in the previous 12 mo (*Figure 5A*); in that same time period, 13% of the town's population moved to another district (*Figure 5B*). The city was also an important in-flow hub in 1991 (*Figure 5C*) and out-flow hub in 2011 (*Figure 5H*). Gaborone was also an important in-flow hub in 1981 and 2001 (*Figure 5A and E*), but only an important out-flow hub in 1991 and 2001 (*Figure 5D and F*).

## Understanding and visualizing the role of migration networks in urbanization

The internal migration networks for each of the four time periods (1981, 1991, 2001, and 2011) are presented in terms of the classification system for our migration-urbanization framework, both as schemata (*Figure 7*) and as Sankey diagrams (*Figure 8*). The framework consists of five classes (see 'Methods'). The class designation (city, town, predominantly urban, partially urban, predominantly rural) of each district over time is provided in *Supplementary file 1*. Both the schemata and the Sankey diagrams show the magnitude of the migratory flows amongst the five classes in the 12 mo before each census. Between 1981 and 2011, the overall percentage of the population living in urban areas increased more than 3.5-fold (from ~18% to ~64%), and the internal migration networks changed substantially.

In 1981, there were only six districts (the two cities and four towns) that were urban areas (*Supplementary file 1a*). The other districts contained several hundred small rural villages and no urbanized areas. Almost all districts had an in-flow and out-flow of migrants: notably, rural-to-urban migratory flows and urban-to-rural counter-flows are apparent (*Figures 7A and 8A*). Taken together, the results show that in 1981 the population was very mobile, the majority of migratory flows were within and amongst rural districts, and rural-to-urban migrations were greater than urban-to-rural counter-flows.

By 1991, due to in situ urbanization, one previously rural district (South East) had become predominantly urban, and four districts (Kweneng East, Kgatleng, Ngamiland South, and Chobe) had become partially urban (*Supplementary file 1b*). Migratory flows appear to be fairly symmetrical (in terms of

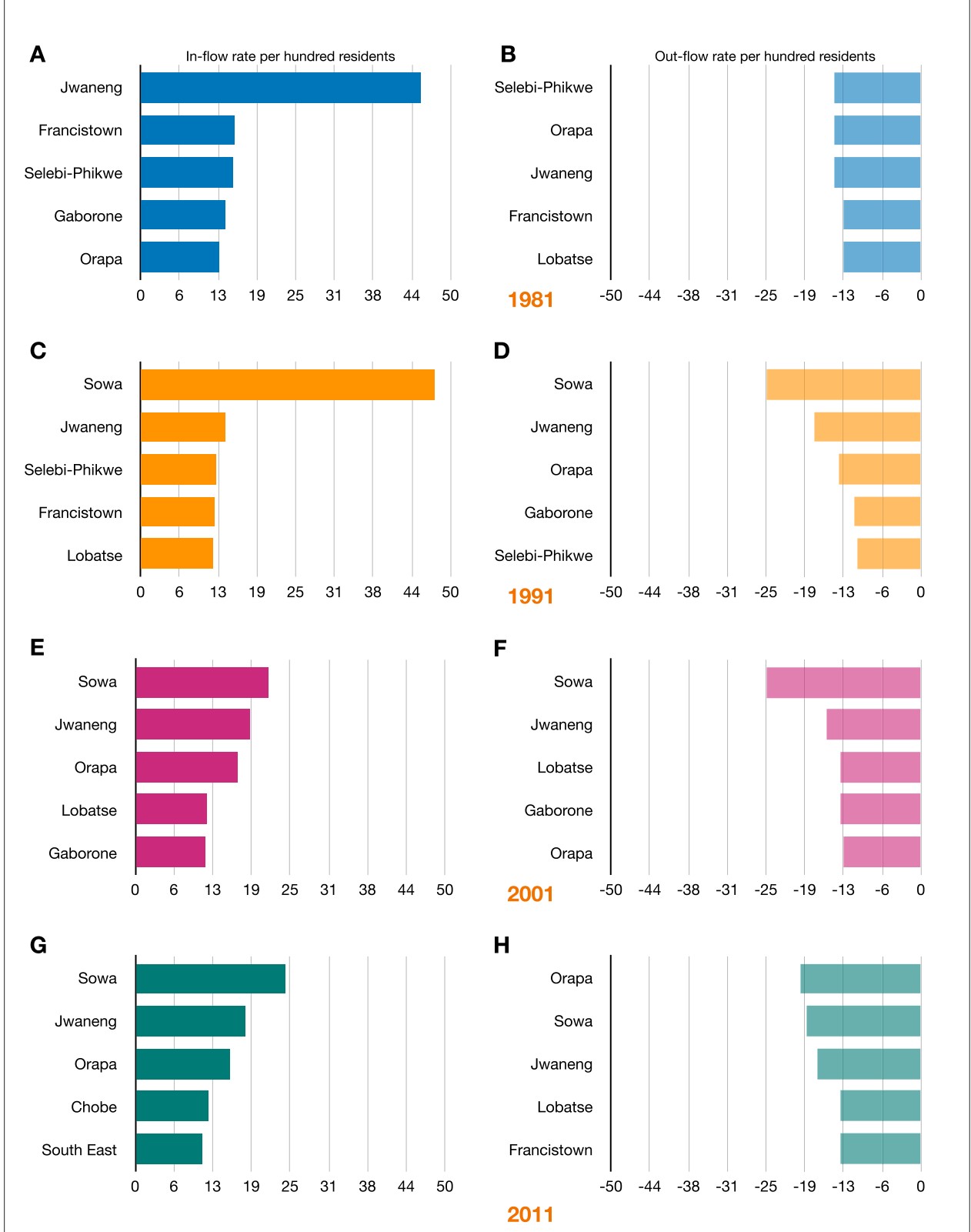

**Figure 5.** The top five in-flow and out-flow migration hubs by census year. In-flow hubs are sorted by the rate per hundred residents of a district's population that moves in from another district; out-flow hubs are sorted by the rate per hundred residents of a district's population that moves out to another district. (**A**) Top in-flow hubs in 1981. (**B**) Top out-flow hubs in 1981. (**C**) Top in-flow hubs in 1991. (**D**) Top out-flow hubs in 1991. (**E**) Top in-flow hubs in 2001. (**F**) Top out-flow hubs in 2001. (**G**) Top in-flow hubs in 2011. (**H**) Top out-flow hubs in 2011.

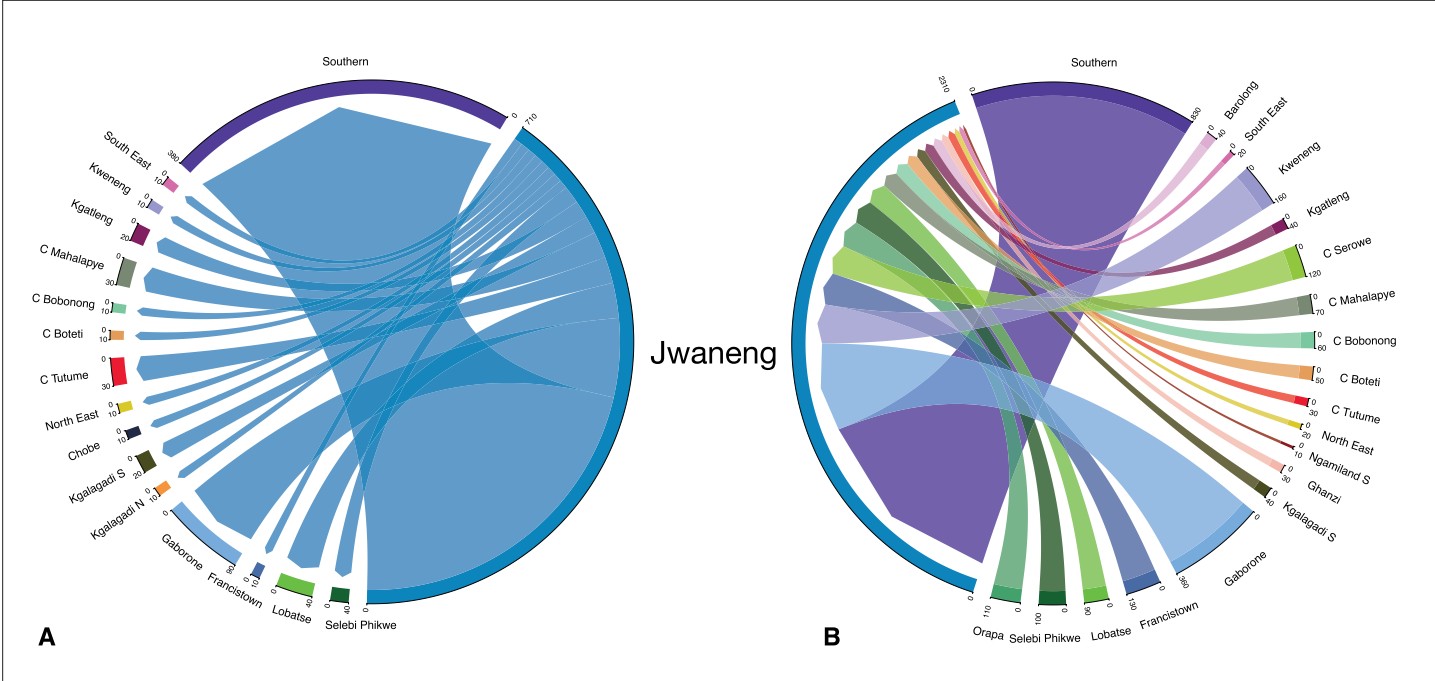

**Figure 6.** Ego network of Jwaneng. The chord diagrams show migrants flowing (**A**) out of, and (**B**) into, the diamond mining town in 1981.

the number of migrants who moved between pairs of districts) throughout the country (*Figure 7B* and *Figure 8B*). The towns had increased 1.5-fold since 1981, and the cities had more than doubled in size. As at the time of the previous census, there were considerable rural-to-urban migratory flows and urban-to-rural counter-flows.

By 2001, considerably more in situ urbanization had occurred: since 1981, four districts had become predominantly urban, six districts partially urban, and only nine districts remained predominantly rural (*Supplementary file 1c*). Notably, some of the migratory flows were now asymmetric: the number of migrants moving from the cities to the predominantly urban and partially urban districts were greater than the migratory counter-flows (*Figure 7C* and *Figure 8C*). The growth rate of towns remained approximately the same as in the previous decade, but the growth rate of the cities had decreased to 1.4-fold; these rates depended upon the number of migrants, births, and deaths.

By 2011, 12 districts had become predominantly or partially urbanized (*Supplementary file 1d*). The growth rate of cities had decreased to 1.2-fold, and the towns had not increased substantially in size. At this time, while most flows were symmetric, migrants were disproportionately leaving the cities in favor of towns and urban villages (*Figure 7D* and *Figure 8D*). Approximately 42% of the population were living in these urban villages; there was at least one urban village in every district.

## The impact of internal migration on the geographic distribution of the population

We evaluated the impact of internal migration on the geographic distribution of the population by calculating two metrics: the Migration Effectiveness Index (MEI) and the Aggregate Net Migration Rate (ANMR). The MEI, which ranges from 0 to 100, quantifies the balance between the migratory flows and counter-flows (*Shryock et al., 1975*). Low values of the MEI indicate that the migratory flows and counter-flows are fairly balanced; the higher the value of the MEI, the greater the asymmetry between flows and counter-flows. The ANMR measures the impact of internal migration on geographically redistributing the population (*Bell et al., 2002*): it identifies the net shift of population between districts per hundred residents per year. The ANMR is determined by both the intensity of migration (as specified by the CMI) and the balance between the migratory flows and counter-flows (as specified by the MEI).

We estimated that the MEI was 7.30 between 1980 and 1981, 7.18 between 1990 and 1991, 5.19 between 2000 and 2001, and 7.77 between 2010 and 2011. These values indicate that the migratory

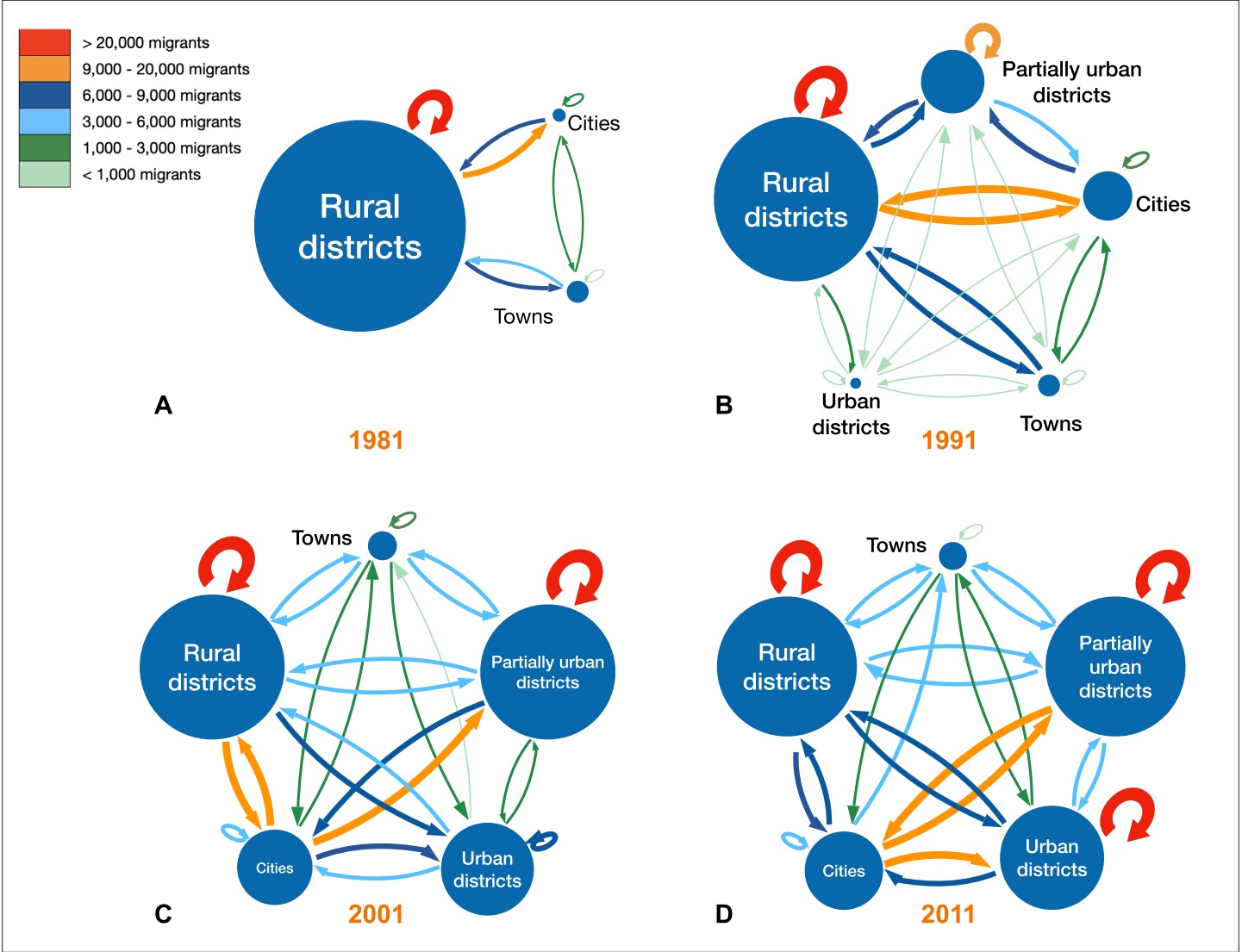

**Figure 7.** Schemata of migration-urbanization framework by census year. (**A**) 1981, (**B**) 1991, (**C**) 2001, and (**D**) 2011. Circles represent the five classes based on an urban/rural classification (see 'Methods'). The radius of each circle is proportional to the number of residents living in the districts in that specific class. The color of each arrow indicates the size of the net migration between classes, as does the thickness of the line: the thicker the line, the greater the number of migrants.

flows and counter-flows in Botswana were fairly symmetric (therefore, internal migration networks were not very effective in geographically redistributing the population) and were not substantially different between each census. We estimated that the ANMR was 0.66 between 1980 and 1981, 0.75 between 1990 and 1991, 0.56 between 2000 and 2001, and 0.80 between 2010 and 2011. Therefore, the overall impact of migration on geographically redistributing the population was very low: the net shift of the population between districts was less than one individual per hundred residents in the 12 mo prior to each census.

## Discussion

Our results support our mobility hypothesis that – during the development of Botswana's generalized HIV epidemic (i.e., between 1981 and 2011) – the population was extremely mobile and the country was highly connected by urban-to-rural migratory flows and rural-to-urban counter-flows. Using our time series of historical data, we found ~10% of the population moved their residency in the 12 mo before each census in 1981, 1991, 2001, and 2011. The constancy of this value at each census suggests

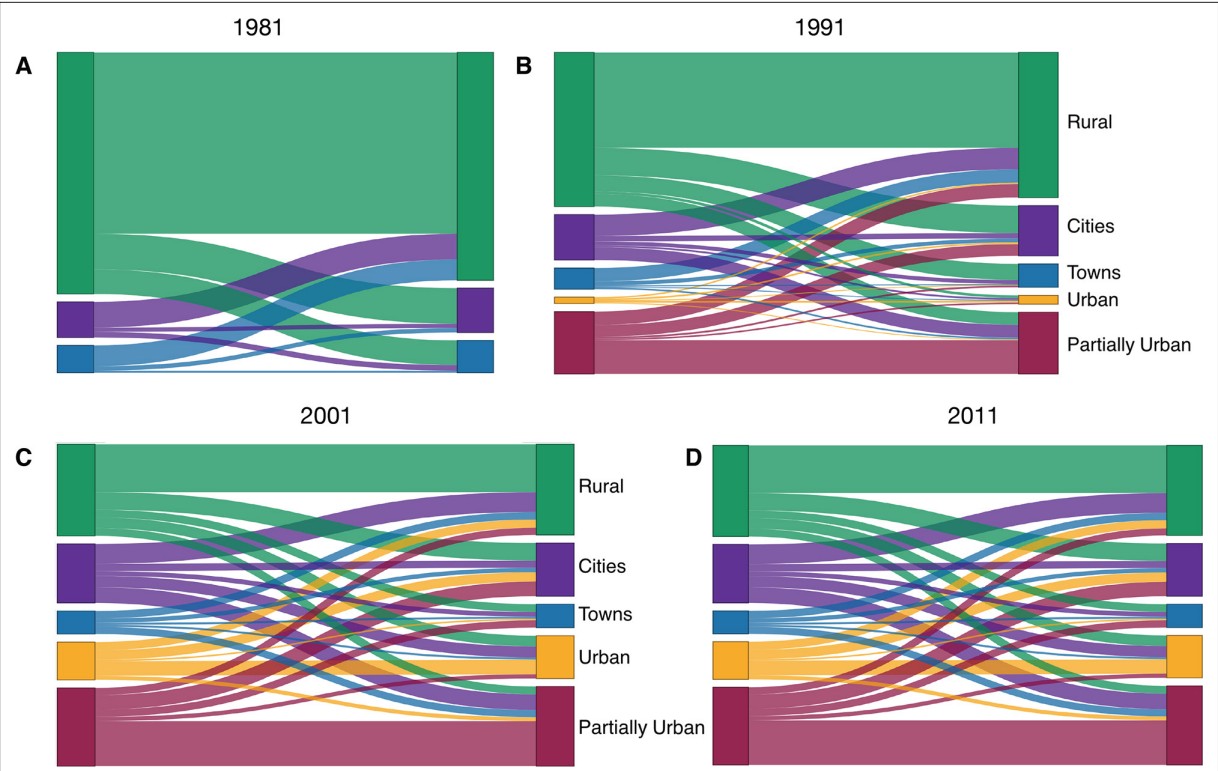

**Figure 8.** Sankey diagrams showing the migration-urbanization framework by census year. (**A**) 1981, (**B**) 1991, (**C**) 2001, and (**D**) 2011. The diagrams show the relative magnitude of migratory flows between the five classes over time. Cities are shown in purple, towns in blue, predominantly urban districts in gold, partially urban districts in magenta, and predominantly rural districts in green.

that the annual migration rate in Botswana was stable at ~10% for each year between 1981 and 2011. Notably, the type of migrants also remained constant over this time period: evenly split by gender, with younger people more likely to migrate. We found that migration occurred within districts, as well as between districts, but that migration between districts was more common. Even in 1981, at the beginning of the period of rapid urbanization, the migration network was highly connected and therefore large-scale population movements linked almost all of the districts in the country. Notably, we found at each census, that although the number of urbanized districts increased (as did their population size) the vast majority of migratory flows between the rural and developing urban districts were balanced by counter-flows.

Our results on migration networks show how a generalized epidemic had the potential to develop: HIV could have diffused, via the migration networks, throughout Botswana. The first cases of AIDS in Botswana were reported in the 1980s (*African Natural Resources Center, 2016*); this suggests that HIV may have been circulating in Botswana in the 1970s, as HIV has an incubation period of 10–12 y (*Hendriks et al., 1993*; *Muñoz et al., 1989*). One phylogenetic study indicates that HIV was introduced into Botswana in the 1960s, but exponential growth did not begin until the 1980s (*Wilkinson et al., 2015*). Another study suggests that HIV spread southward from Kinshasa (the epicenter of the HIV epidemic) in the Democratic Republic of the Congo via travelers on the railroad that was built in the 1800s (*Faria et al., 2014*). Notably, this railroad goes through Francistown, which is one of Botswana's two cities, and where HIV prevalence quickly rose to high levels in the early 1990s (*UNAIDS and World Health Organization, 2004*). It is possible that HIV was introduced into the mining towns of Botswana from overseas or South Africa. In the 1970s and 1980s, many young men from Botswana went to work in the mines in South Africa, returning as the mining industry became established in their home country. Our results show that, by 1981, the migration networks (for women and men) linked almost every district in Botswana. This indicates that HIV-infected migrants from Francistown and/or the mining towns (where AIDS cases were first reported and HIV prevalence quickly rose to high levels) could have seeded HIV sub-epidemics in multiple rural districts throughout the country. HIV could

then have spread widely within these districts due to the high rates of within-district migration that we have found. Due to the high and continuous migration rates, the sub-epidemics in rural districts were likely to have been seeded numerous times by HIV-infected migrants. When HIV was first introduced into rural districts, the sub-epidemics in these districts may have been maintained by source-sink dynamics (*Okano et al., 2020*): districts where transmission was high enough to be self-sustaining (e.g., mining towns) could have maintained sub-epidemics in rural districts where transmission was too low to be self-sustaining. Our results suggest that migratory flows, and counter-flows, between high and low prevalence areas (e.g., between the mining towns and rural areas, or between Botswana's two cities and rural areas) are likely to have existed, at least from the early 1980s onward. These flows would have functioned as 'transmission corridors' (*Okano et al., 2021*), and created high-risk flows that crossed the country (*Valdano et al., 2021*), that is, HIV-infected people migrating from high-prevalence to lower-prevalence areas, and uninfected people migrating from low-prevalence areas to higher-prevalence areas. Taken together, our results show that the highly connected migration networks, with high migratory flows and counter-flows, could have contributed to the development of Botswana's generalized epidemic by distributing HIV throughout the country in the general population.

Our results on migration networks potentially explain how Botswana's HIV epidemic became hyperendemic. Migrants have been shown to be at high risk of acquiring HIV infection (*Camlin and Charlebois, 2019*; *Dobra et al., 2017*; *Olawore et al., 2018*); our results suggest that, due to migration alone, each year ~10% of the population of Botswana was at high risk. The migrants in Botswana were young women and men, a group generally at the highest risk of HIV infection through sexual transmission. Mining towns and cities would have been transmission hot-spots (for both women and men) due to high levels of risky sexual behavior (as evidenced by high HIV prevalence; *Barnett et al., 2002*; *UNAIDS and World Health Organization, 2004*) and a high probability of encountering an HIV-infected individual as a sex partner. Due to the high prevalence of HIV, high turnover rates, and a high level of connectivity to other districts, the mining towns and cities could have functioned as geographically defined core groups for sexual transmission (for both women and men). Some individuals could have been part of a high-risk core group when they were living in the cities and mining towns, and decreased their risky behavior when they returned to the rural districts. For example, FSWs in Botswana can have an extremely high number of sex partners in mining towns and urban areas (~7 per week) and remain as FSWs for ~4 y (*Ministry of Health Botswana, 2013*). Notably we found that the mining towns and cities were both migration in-flow hubs and migration out-flow hubs. The fact that they were migration in-flow hubs would have led to a high in-flow of uninfected individuals from rural districts where prevalence was low; the fact that they were migration out-flow hubs (and had a very high prevalence of HIV) would have led to a high out-flow of HIV-infected individuals that could have seeded, and reseeded, sub-epidemics in rural districts. Therefore, rural-to-urban migration coupled with urban-to-rural migration could have been a very important driver in the Botswana epidemic becoming hyperendemic. Without it, prevalence could have reached very high levels in the mining towns and cities, but remained fairly low in rural districts.

We have discussed the potential impact of mobility on Botswana's HIV epidemic, and focused on migration, that is, one directional movement in terms of a permanent relocation of residency. This type of migration, by changing an individual's sexual network and social environment, has been shown to increase the risk of acquiring HIV for both women and men (*Anglewicz et al., 2018*; *Camlin and Charlebois, 2019*; *Dzomba et al., 2019*; *Low et al., 2021*; *Olawore et al., 2018*; *Thorp et al., 2023*). Short-term mobility (e.g., short-term circular migration, where the trip can range in duration from overnight to an entire season) can also affect HIV transmission dynamics. Circular migrants have been shown to both have an increased risk of acquiring HIV, and of transmitting HIV (*Camlin and Charlebois, 2019*; *Palk and Blower, 2015*). The greater the number of trips and/or the duration of the trip, the greater the risk (*Palk and Blower, 2015*). We note that both migration and short-term mobility are important, and their relative importance to each other is likely to evolve over time as a generalized HIV epidemic diffuses through the population. Their relative importance is also likely to vary amongst countries in sub-Saharan Africa.

Our results also provide insights into the role of Botswana's internal migration network during a period of rapid urbanization. The urbanization process was first studied quantitatively by Ravenstein, who analyzed the 1871 and 1881 censuses of Great Britain and Ireland, and calculated the flows of

lifetime migrants to urban centers (*Ravenstein, 1885*). He showed that internal migration from rural areas was essential to the growth of industrial cities and towns. Additionally, he demonstrated that migration was extremely important in changing settlement patterns and geographically redistributing the population in both Great Britain and Ireland. In contrast, we found at each census, that generally the majority of rural-to-urban migratory flows were balanced by counter-flows from urban areas to rural villages. Consequently, even though there was a very high migration rate, we found that migration did not drive urbanization, and also had relatively little impact on geographically redistributing Botswana's population. Differences between Botswana and high-income countries, in the role of internal migration on urbanization, were due to in situ urbanization occurring in Botswana (rather than the rapid growth of cities and towns as in high-income countries) and high rural-to-urban migratory flows being balanced by high urban-to-rural counter-flows. During the 30-year time period that we have analyzed (1981–2011), population growth in Botswana was high: the population doubled. Notably, the Organization for Economic Co-operation and Development recently reported that since 1990, in situ urbanization and high population growth rates have driven Africa's rapid growth in urbanization (*Moriconi-Ebrard et al., 2020*). Countries are at different stages of urbanization, but urbanization is occurring in essentially all African countries (*United Nations, Department of Economic and Social Affairs, & Population Division, 2019*).

Our results on the role of migration on urbanization have significant implications for predicting migratory flows in Africa. Africa has high migration rates and is one of the least urbanized places in the world (*United Nations, Department of Economic and Social Affairs, & Population Division, 2019*): the number of urban residents on the continent approximately doubled between 1995 and 2015, and is projected to double again by 2040 (*Lall et al., 2017*). Consequently, it is important to be able to predict migration patterns and flows in Africa (*Okano et al., 2021*); for example, for urban planning and developing efficient resource allocation strategies. The mathematical models currently being used to predict migratory flows in Africa (*Ciavarella and Ferguson, 2021*; *Marshall et al., 2018*; *Meredith et al., 2021*) have been developed based on migratory processes that have been observed, and census data that have been collected, in high-income countries. The two most commonly used predictive models, the gravity model (*Stewart, 1948*) and the radiation model (*Simini et al., 2012*), make the central assumption that individuals are attracted to move to a new location based on the population size at the destination relative to the size at the origin, moving from the smaller to the larger population. Under this assumption, it is expected that we would have found asymmetry in migration flows between rural-to-urban and urban-to-rural areas, and a migration-induced geographic redistribution of the population. However, we found almost symmetric migratory counter-flows occurring throughout the country, and that migration had almost no impact on changing the geographic distribution of the population. Therefore, our results suggest that new mathematical models are needed to predict migration in Botswana, and – potentially – in other rapidly urbanizing countries in Africa where in situ urbanization is occurring.

Our study has several limitations. First, we have examined only internal migration (i.e., migration that takes place within a country) and have not included international migration. However, for the three-decade time period that we have investigated, the number of international migrants to Botswana was much smaller than the number of internal migrants (*Statistics Botswana, 2014*; *Statistics Botswana, 2015*). Second, all of our analyses are focused on Botswana in order to analyze the impact of migration networks on the development of a generalized hyperendemic epidemic; we have not evaluated migration networks in other African countries. We recommend that studies, such as we have conducted here, are conducted in these other countries by analyzing a time series of historical micro-census data, especially in other countries in sub-Saharan Africa that have generalized hyperendemic HIV epidemics, such as Eswatini and Lesotho. Third, a potential limitation is that the IPUMS-International (Integrated Public Use Microdata Series-International) data that we used are samples rather than the complete censuses; we used this approach as it is generally not possible to obtain complete census data due to privacy reasons. Fourth, while we have gained insights into understanding the development of the generalized hyperendemic HIV epidemic in Botswana, we have not conducted an analysis of the transmission dynamics of the epidemic in order to explore these insights quantitatively. This is the focus of current research in which we are analyzing a geospatial mathematical model of the epidemiological evolution of the HIV epidemic in Botswana. Finally, we

have examined only one process (internal migration) that has affected the evolution of Botswana's epidemic; other factors could also have been important.

All HIV epidemics in SSA are generalized; HIV can only diffuse through a population by the movement of people. To our knowledge, our study is the first to study population-level mobility patterns and rates in any country in sub-Saharan Africa. Taken together, our results identify particular characteristics of migration networks in Botswana that could explain why HIV prevalence rose to very high levels in many districts throughout the country. Migration networks in other countries with generalized HIV epidemics may be very different. If mining towns and/or cities in these countries were not migratory in-flow and out-flow hubs with high turnover of their populations, as in Botswana, they would not have served as major sources of infection for dispersing HIV throughout the general population. Botswana has recently achieved the 95-95-95 targets (*Mine et al., 2022*) that UNAIDS specified needed to be reached by 2030 in order to eliminate HIV: 95% of people living with HIV need to be diagnosed, 95% of the diagnosed need to be on treatment, and 95% of those on treatment need to be virally suppressed. However, transmission is continuing (*Magosi et al., 2022*), as is urbanization and large-scale population movements (*Okano et al., 2021*). The same processes that we have identified that may have contributed to the development of the generalized hyperendemic HIV epidemic in Botswana may prevent the elimination of HIV.

## Methods
### Data
To conduct our analyses, we used representative samples of micro-census data extracted from the IPUMS-International database (*Minnesota Population, 2020*): these data are anonymized individual-level data (*Ruggles et al., 2015*). IPUMS-International currently disseminates data from 547 censuses and surveys in 103 countries worldwide (*Minnesota Population, 2020*). The Botswana dataset consists of representative 10% samples from the original censuses and includes anonymized individual-level data on age, gender, and residence (current, as well as 12 mo prior). The data also includes individual survey weights that allow for population-level estimation (*Ruggles et al., 2015*).

To examine historical trends in internal migration networks and urbanization, we analyzed data collected in the 1981, 1991, 2001, and 2011 censuses. We used data on internal migration that had occurred in the 12 mo prior to each census: an individual was classified as a migrant if they had changed their permanent residency within that 1-year interval.

### Estimating the incidence of internal migration
We estimated the incidence of internal migration by calculating the CMI. This statistic represents the overall incidence, or level of internal migration (between district plus within district), per hundred residents over a year. The mathematical definition of the CMI is given in *Equation (1)*:

$$CMI = 100 \cdot \frac{M}{P} \tag{1}$$

where $M$ is the total number of internal migrants and $P$ is the population size of Botswana in a given year. $M$ is calculated as either the total in-flow of migrants into all of the districts or the total out-flow of migrants from all of the districts. These two quantities are equivalent as the in-flow of migrants to any district results from an out-flow of migrants from one or more other districts. Therefore, $M = \sum_i D_i = \sum_i O_i$, where $D_i$ denotes the number of migrants who move into each district $i$, and $O_i$ denotes the number of migrants who move out of each district $i$.

### Constructing gender-stratified age structure pyramids
We aggregated the migration data from each census by gender and age (using 5-year age groupings) to construct population pyramids; these pyramids show the age-gender demographics of all individuals internally migrating in the 12 mo prior to each census.

### Calculating migratory flows
We define a district-level migratory flow as the number of migrants who change their residency from one district to another during the 12 mo prior to the census. We calculated migratory flows between

each pair of districts. The country consists of 28 administrative districts (*Okano et al., 2021*). Each city and town are separate administrative districts.

## Calculating annual turnover rates

We defined the annual turnover rate for a district as the net change in its annual migration rate per hundred residents.

$$TO_i = 100 \cdot \frac{\left[D_i - O_i\right]}{P_i} \tag{2}$$

Here the turnover rate $TO_i$ for district $i$ is a function of the number of in-migrants $D_i$ and out-migrants $O_i$ within the past year, and its population $P_i$ at the beginning of the year.

## Reconstructing migration networks

We used the micro-census data to construct Origin-Destination (OD) matrices: the origin was the district that an individual lived in 12 mo prior to the census, the destination was the district they lived in at the time of the census. Coefficients of these matrices specify the number of migrants who moved between each pair of districts in the 12 mo prior to each census. We defined the within district migration intensity (WDMI) for each district as the number of internal movements per hundred residents.

## Identifying in-flow and out-flow migration hubs

Migration hubs are those districts where recent migration has a sizeable impact on the size of the resident population by either bringing it down (out-flow hubs) or increasing it (in-flow hubs) above the average. In-flow migration hubs were identified by calculating the total number of in-migrants to a node/district and dividing by the district's population that year. Out-flow migration hubs were identified by calculating the total number of out-migrants to a node/district and dividing by the district's population that year. We list the top five in-flow and out-flow hubs for each census year.

## Understanding and visualizing the role of migration networks in urbanization

To understand, and visualize, the role of migration networks in urbanization, we developed a classification system for a migration-urbanization framework. The framework consists of five classes that are defined (at any point in time) based on the degree of urbanization of the district at the time of the most recent census: (i) predominantly rural (<40% of the population live in urban areas), (ii) partially urban (40–60% of the population live in urban areas), (iii) predominantly urban (>60% of the population live in urban areas), (iv) town (100% of the population live in urban areas), or (v) city (100% of the population live in urban areas). Classes (i)–(iii) only include urban areas that develop by in situ urbanization, that is, rural villages transforming into urban villages. There are migratory flows between the five classes, and flows within each of the five classes: therefore, 25 migratory flows are possible.

The migration-urbanization framework can be visualized in two formats: (i) a schematic figure showing the five classes, the number of residents in each class, and migratory flows (and counter-flows) amongst and within the classes; and (ii) Sankey diagrams showing the number of migrants who move within, and between, the five classes.

We used the micro-census data to parameterize the framework for Botswana. Specifically, we estimated – at the time of each census – the population size of each district and determined how many individuals lived in rural and urban areas. We used these estimates to classify all districts into one of the five classes. We then calculated the migratory flows and counter-flows between, and within, districts in the 12 mo prior to each survey. These parameter estimates are given for the 1981 census (*Supplementary file 1a*), the 1991 census (*Supplementary file 1b*), the 2001 census (*Supplementary file 1c*), and the 2011 census (*Supplementary file 1d*). Between each census, districts change in size due to births, deaths and migration. Also, between each census, districts can be reclassified based upon their increased level of urbanization.

## Evaluating the impact of internal migration on the geographic distribution of the population

We evaluated the impact of internal migration on geographically redistributing the population by calculating two metrics: the MEI and the ANMR. The MEI indicates the effectiveness (or efficiency) of migration as a mechanism for population redistribution. The mathematical definition of the MEI is given in *Equation (3)*:

$$MEI = 100 \cdot 0.5 \sum_i \frac{|D_i - O_i|}{M} \tag{3}$$

The ANMR measures the impact of migration on population redistribution; it identifies the net shift of the population between regions per hundred residents in the country. The mathematical definition of the ANMR is given in *Equation (4)*:

$$ANMR = CMI \cdot \frac{MEI}{100} \tag{4}$$

## Acknowledgements

We acknowledge the Central Statistics Office (CSO) Botswana for collecting the data archived by IPUMS. We are grateful to Nelson Freimer for discussions throughout the course of this research. JS, JTO, JP, and SB acknowledge the financial support of the National Institute of Allergy and Infectious Diseases, National Institutes of Health grants R56 AI152759 and R01 AI167713.

## Additional information

### Funding

| Funder | Grant reference number | Author |
| --- | --- | --- |
| National Institute of Allergy and Infectious Diseases | R56 AI152759 | Janet Song<br>Justin T Okano<br>Joan Ponce<br>Sally Blower |
| National Institute of Allergy and Infectious Diseases | R01 AI167713 | Janet Song<br>Justin T Okano<br>Joan Ponce<br>Sally Blower |

The funders had no role in study design, data collection and interpretation, or the decision to submit the work for publication.

### Author contributions

Janet Song, Justin T Okano, Software, Formal analysis, Validation, Methodology, Writing - review and editing; Joan Ponce, Software, Formal analysis, Visualization, Methodology, Writing - review and editing; Lesego Busang, Khumo Seipone, Conceptualization, Writing - review and editing; Eugenio Valdano, Methodology, Writing - review and editing; Sally Blower, Conceptualization, Supervision, Funding acquisition, Writing - original draft, Writing - review and editing

### Author ORCIDs

Sally Blower http://orcid.org/0000-0003-4342-3911

### Decision letter and Author response

Decision letter https://doi.org/10.7554/eLife.85435.sa1
Author response https://doi.org/10.7554/eLife.85435.sa2

## Additional files

### Supplementary files

• Supplementary file 1. Tables of migration metrics by district for (a) 1980–1981, (b) 1990–1991, (c) 2000–2001, and (d) 2010–2011. These tables present each district's urban/rural classification, population size, total number of migrants (both within and between districts), within district migration intensity per hundred residents (WDMI), and population turnover per hundred residents. The urban/rural classes were city (C), town (T), predominantly urban (U), partially urban (PU), and predominantly rural (R). Population sizes are tabulated for residents for whom there was 1-year migration data. In 2010–2011, the Central Kgalagadi Game Reserve district is denoted CKGR.

• MDAR checklist

### Data availability

All data needed to evaluate the conclusions in the paper are presented in the paper, or freely available for registered users at the IPUMS International website: https://doi.org/10.18128/D020.V7.3. We note that we did not collect these data, nor are they permitted to be posted to other repositories. All code needed to reproduce all parts of this analysis are available from the first author's GitHub page: https://github.com/janetsong80/pop-mobility-botswana-hiv (copy archived at: https://zenodo.org/badge/latestdoi/594401539).

The following previously published dataset was used:

| Author(s) | Year | Dataset title | Dataset URL | Database and Identifier |
|---|---|---|---|---|
| Minnesota Population Center | 2020 | Integrated Public Use Microdata Series, International | https://doi.org/10.18128/D020.V7.3 | IPUMS: Version 7.3, 10.18128/D020.V7.3 |

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
