## [Editor Report]

This valuable paper uses representative samples of micro-census data from Botswana to describe migration rates over four points in time, from 1981 to 2011. The authors use compelling descriptive data to present migration characteristics where roughly 10% of the population moved in the past year – with equal numbers of men and women, and with migration between districts more common than within districts. Preliminary data indicated migration patterns could have supported HIV diffusion, this can be a starting point for more in-depth analyses. The work will be of interest to those studying human movement and its impact on diseases.

---

## [Decision Letter]

**Decision letter after peer review:**

Thank you for submitting your article "Population mobility and the development of Botswana's generalized HIV epidemic: a network analysis" for consideration by *eLife*. Your article has been reviewed by 2 peer reviewers, and the evaluation has been overseen by a Reviewing Editor and Bavesh Kana as the Senior Editor. The following individual involved in the review of your submission has agreed to reveal their identity: Nick Ruktanonchoi (Reviewer #2).

Essential revisions:

As a general statement, reviewers felt the conclusions were insufficiently linked to HIV-related outcomes and metrics. Please consider this and adjust the text accordingly. More detail on these sentiments is available in the individual reviewer comments. Other issues that need to be addressed are detailed below.

1. A limitation of the paper was that very little context, outside of migration rates, was provided. Is there any additional information about economic growth, or political events for example, that could clarify or add context to these migration flows? As it stands now, these analyses are quite basic and don't take into account underlying demographic, economic, or political trends that compel people to move.

2. Throughout the paper, it is alluded to Botswana having high rates of population movement to urban areas. To some degree, this is self-evident, but it would be useful if there are any other countries/periods to compare against. Is 10 persons per 100 a lot compared to other countries?

3. It would be useful to include some text discussing why migratory mobility might be more relevant for HIV than short-term trips, or more discussion of short-term trips in general. For example, the FSM average stay-time being 4 years seems like a compelling reason to think about migration (it happens frequently enough that it could be a significant contributor to new cases elsewhere).

4. In general, overlaying HIV risk with some of the outcomes would be interesting. Are there data on the age distribution of HIV cases during 1980-2000? It would be interesting to see that overlaid on the figure showing the demographics of migrants over time. (Figure 1)

5. Maps would help visualize some of the other results. For example, it would be interesting to see some of the Turnover/WDMI metrics by district on a map instead of in a table. It would also reduce the number of tables. As the manuscript currently stands, Tables 1 – 4 feel more appropriate for supplementary information rather than 4 separate tables in the main text.

6. Turnover in particular would be nice to see on a map/slightly more emphasized. A lot of the metrics are fairly similar in both directions (as the authors point out in the discussion), but it would be additionally useful to see the net flow for some of the metrics.

7. Figure 5 might be better served with population flow reflected as arrow size rather than color. Right now it's hard to tell if there are any obvious trends.

8. Figure 1 – This would be a nice place to add HIV context, if there is information on the age structure of HIV cases.

9. Figure 3: consider putting them all on the same scale, so differences over time are clearer.

10. Add citation to the statement: "For example, migrants in other African countries tend to have a very male-based sex ratio." Is this true?

11. Typo in the equation on top of page 21, line 437.

*Reviewer #1 (Recommendations for the authors):*

Thank you for the opportunity to review "Population mobility and the development of Botswana's generalized HIV epidemic: a network analysis." I found the data presented in the paper about the historical qualities and characteristics of migration to be clear. But I also felt like this paper lacked significant depth. These data are nonetheless useful to present: but the statements made in the paper, linking these data to the HIV epidemic, feel overstated.

One other limitation of the paper was that very little context, outside of migration rates, was provided. Is there any additional information about economic growth, or political event for example, that could clarify or add context to these migration flows? As it stands now, these analyses are quite basic and don't take into account underlying demographic, economic or political trends.

Figure 3: consider putting them all on the same scale, so differences over time are clearer.

Add citation to the statement: "For example, migrants in other African countries tend to have a very male-based sex ratio." Is this true?

Typo in the equation on top of page 21, line 437.

In conclusion, I find this descriptive paper to be clear, but not particularly impactful. It may only be of interest to a small subset of readers. As mentioned before, these data can provide the base of a number of additional analyses.

*Reviewer #2 (Recommendations for the authors):*

Broadly I thought this was an interesting paper, but could go a little further in terms of relation to the HIV epidemic itself, and some more context generally:

– Throughout the paper, it is alluded to Botswana having high rates of population movement to urban areas. To some degree I think the evidence shows this self-evidently, but it would be useful if there are any other countries/periods to compare against. Is 10 persons per 100 a lot compared to other countries?

– It would be useful to include some text discussing why migratory mobility might be more relevant for HIV than short-term trips, or more discussion of short-term trips in general. For example, the FSM average stay-time being 4 years seems like a compelling reason to think about migration (it happens frequently enough that it could be a significant contributor to new cases elsewhere).

– In general, overlaying HIV risk with some of the outcomes would be interesting. Are there data on the age distribution of HIV cases during 1980-2000? It would be interesting to see that overlaid on the figure showing the demographics of migrants over time. (Figure 1)

– Maps would help visualize some of the other results. For example, it would be interesting to see some of the Turnover/WDMI metrics by district on a map instead of in a table. It would also reduce the number of tables--I think right now Tables 1 – 4 feel more appropriate for SI than 4 separate tables in the main text.

– Turnover in particular would be nice to see on a map/slightly more emphasized. A lot of the metrics are fairly similar in both directions (as the authors point out in the discussion), but it would be additionally useful to see the net flow for some of the metrics.

– Figure 5 might be better served with population flow reflected as arrow size rather than color. Right now it's hard to tell if there are any obvious trends.

– Figure 1 – This would be a nice place to add HIV context if there is information on the age structure of HIV cases.

---

## [Author Response]

Essential revisions:As a general statement, reviewers felt the conclusions were insufficiently linked to HIV-related outcomes and metrics. Please consider this and adjust the text accordingly. More detail on these sentiments is available in the individual reviewer comments. Other issues that need to be addressed are detailed below.

We have now rewritten the abstract (see highlighted sentences) and discussion (lines 302-377) in order to link our conclusions regarding migration networks and metrics more closely to HIV-related outcomes. We have also included an additional Figure (Figure 1): this is a district-level map of Botswana that shows the countrywide geographic variation in HIV prevalence (the map is based on population-level HIV-testing data that were collected in 2008).

1. A limitation of the paper was that very little context, outside of migration rates, was provided. Is there any additional information about economic growth, or political events for example, that could clarify or add context to these migration flows? As it stands now, these analyses are quite basic and don't take into account underlying demographic, economic, or political trends that compel people to move.

In response to this concern we have expanded the text in the introduction to provide more context regarding political, demographic and economic factors (Introduction: lines 66-75). We have also expanded our discussion of the implications of our results (and of additional results that we have included: lines 263-283) for understanding the role of internal migration on urbanization in Botswana (Discussion: lines 379-420); urbanization occurred simultaneously to the development of Botswana’s generalized hyperendemic HIV epidemic.

2. Throughout the paper, it is alluded to Botswana having high rates of population movement to urban areas. To some degree, this is self-evident, but it would be useful if there are any other countries/periods to compare against. Is 10 persons per 100 a lot compared to other countries?

The purpose of our analysis is to characterize the situation in Botswana between 1981 and 2011. We agree with the Reviewer that it is self-evident that the rates of migration that we find are high. Unfortunately – to the best of our knowledge – there do not appear to be any similar analyses of historical migration data, over the same time period and at the similar administrative level, in other countries that we can compare our migration rates with. Calculating comparable rates is a complex problem because: (i) very few censuses collect data on short-term population movement, and (ii) there is substantial heterogeneity in the geographic size of administrative units (which is necessary in order to make meaningful comparisons). For instance, of countries near Botswana, only Namibia also had a census that tracked one-year movements. While the rate of movement is smaller (2% in 2006) than Botswana’s, this may be in part due to Namibia’s regions being geographically larger (and hence requiring more travel to traverse) than Botswana’s districts.

3. It would be useful to include some text discussing why migratory mobility might be more relevant for HIV than short-term trips, or more discussion of short-term trips in general. For example, the FSM average stay-time being 4 years seems like a compelling reason to think about migration (it happens frequently enough that it could be a significant contributor to new cases elsewhere).

We have added the paragraph (given below) to the revised text (Discussion: lines 364-377). In the text, we have included citations for the paragraph.

“We have discussed the potential impact of mobility on Botswana’s HIV epidemic, and focused on migration, i.e., one directional movement in terms of a permanent re-location of residency. This type of migration, by changing an individual’s sexual network and social environment, has been shown to increase the risk of acquiring HIV for both women and men. Short-term mobility (e.g., short-term circular migration, where the trip can range in duration from overnight to an entire season) can also affect HIV transmission dynamics. Circular migrants have been shown to both have an increased risk of acquiring HIV, and of transmitting HIV. The greater the number of trips and/or the duration of the trip, the greater the risk. We note that both migration and short-term mobility are important, and their relative importance to each other is likely to evolve over time as a generalized HIV epidemic diffuses through the population. Their relative importance is also likely to vary amongst countries in sub-Saharan Africa.”

4. In general, overlaying HIV risk with some of the outcomes would be interesting. Are there data on the age distribution of HIV cases during 1980-2000? It would be interesting to see that overlaid on the figure showing the demographics of migrants over time. (Figure 1)

We agree that overlaying HIV risk with some of the other outcomes would be interesting. However, unfortunately, there are not any data on the age distribution of HIV cases during 1980-2000. Such data were not collected in Botswana for the earliest AIDS cases in the interest of maintaining anonymity due to concerns regarding stigma.

However, we have added to the revised manuscript (Introduction: lines 97-99) that the first reported AIDS cases were young men who were miners. We have added a citation to support this statement.

5. Maps would help visualize some of the other results. For example, it would be interesting to see some of the Turnover/WDMI metrics by district on a map instead of in a table. It would also reduce the number of tables. As the manuscript currently stands, Tables 1 – 4 feel more appropriate for supplementary information rather than 4 separate tables in the main text.

We agree with the Reviewer, and have now included maps of Net flow (Figure 3), Turnover (Figure 3—figure supplement 1) and WDMI (Figure 3—figure supplement 2) over time. As suggested, we have now included Tables 1-4 as supplementary file 1.

6. Turnover in particular would be nice to see on a map/slightly more emphasized. A lot of the metrics are fairly similar in both directions (as the authors point out in the discussion), but it would be additionally useful to see the net flow for some of the metrics.

We have now included maps of Turnover (Figure 3—figure supplement 1) over time, as well as net flow (Figure 3).

7. Figure 5 might be better served with population flow reflected as arrow size rather than color. Right now it's hard to tell if there are any obvious trends.

To increase the clarity of this figure, we now use both arrow size and color – this is now Figure 7.

8. Figure 1 – This would be a nice place to add HIV context, if there is information on the age structure of HIV cases.

Please see our response to point 4 in “Essential revisions” section.

9. Figure 3: consider putting them all on the same scale, so differences over time are clearer.

We have now done this – this is now Figure 5.

10. Add citation to the statement: "For example, migrants in other African countries tend to have a very male-based sex ratio." Is this true?

The answer is complex, as it depends on how migration is defined and the historical period that is being referred to. Therefore, we have now removed this statement.

11. Typo in the equation on top of page 21, line 437.

We apologize that this equation was confusing. It does not contain a typo. The total number of internal migrants, *M*, can be calculated as either (i) the total number of in-flow migrants into all of the districts, or (ii) the total number of out-flow migrants from all of the districts. These quantities are equivalent. This relationship has to hold as the out-flow of migrants from any district is the in-flow of migrants into one or more other districts. We have now clarified this in the text (Methods: lines 481-485).

Reviewer #1 (Recommendations for the authors):Thank you for the opportunity to review "Population mobility and the development of Botswana's generalized HIV epidemic: a network analysis." I found the data presented in the paper about the historical qualities and characteristics of migration to be clear. But I also felt like this paper lacked significant depth. These data are nonetheless useful to present: but the statements made in the paper, linking these data to the HIV epidemic, feel overstated.

In response, in the discussion (lines 302-362) we clarify how the migration networks that we have identified may have impacted the development of the generalized hyperendemic HIV epidemic in Botswana.

One other limitation of the paper was that very little context, outside of migration rates, was provided. Is there any additional information about economic growth, or political event for example, that could clarify or add context to these migration flows? As it stands now, these analyses are quite basic and don't take into account underlying demographic, economic or political trends.

Please see our response to point 1 in “Essential revisions” section.

Figure 3: consider putting them all on the same scale, so differences over time are clearer.

We have now done this – this is now Figure 5.

Add citation to the statement: "For example, migrants in other African countries tend to have a very male-based sex ratio." Is this true?

The answer is complex, as it depends on how migration is defined and the historical period that is being referred to. Therefore, we have now removed this statement.

Typo in the equation on top of page 21, line 437.

Please see our response to point 11 in “Essential revisions” section.

Reviewer #2 (Recommendations for the authors):Broadly I thought this was an interesting paper, but could go a little further in terms of relation to the HIV epidemic itself, and some more context generally:– Throughout the paper, it is alluded to Botswana having high rates of population movement to urban areas. To some degree I think the evidence shows this self-evidently, but it would be useful if there are any other countries/periods to compare against. Is 10 persons per 100 a lot compared to other countries?

Please see our response to point 2 in “Essential revisions” section.

– It would be useful to include some text discussing why migratory mobility might be more relevant for HIV than short-term trips, or more discussion of short-term trips in general. For example, the FSM average stay-time being 4 years seems like a compelling reason to think about migration (it happens frequently enough that it could be a significant contributor to new cases elsewhere).

Please see our response to point 3 in “Essential revisions” section.

– In general, overlaying HIV risk with some of the outcomes would be interesting. Are there data on the age distribution of HIV cases during 1980-2000? It would be interesting to see that overlaid on the figure showing the demographics of migrants over time. (Figure 1)

Please see our response to point 4 in “Essential revisions” section.

– Maps would help visualize some of the other results. For example, it would be interesting to see some of the Turnover/WDMI metrics by district on a map instead of in a table. It would also reduce the number of tables--I think right now Tables 1 – 4 feel more appropriate for SI than 4 separate tables in the main text.

We agree with the Reviewer, and have now included maps of Net flow (Figure 3), Turnover (Figure 3—figure supplement 1) and WDMI (Figure 3—figure supplement 2) over time. As suggested, we have now included Tables 1 – 4 as supplementary file 1.

– Turnover in particular would be nice to see on a map/slightly more emphasized. A lot of the metrics are fairly similar in both directions (as the authors point out in the discussion), but it would be additionally useful to see the net flow for some of the metrics.

We have now included maps of Turnover (Figure 3—figure supplement 1) over time, as well as net flow (Figure 3).

– Figure 5 might be better served with population flow reflected as arrow size rather than color. Right now it's hard to tell if there are any obvious trends.

To increase the clarity of this figure, we now use both arrow size and color – this is now Figure 7.

– Figure 1 – This would be a nice place to add HIV context if there is information on the age structure of HIV cases.

Please see our response to point 4 in “Essential revisions” section.